# Evaluation of Air Contamination in Orthopaedic Operating Theatres in Hospitals in Southern Italy: The IMPACT Project

**DOI:** 10.3390/ijerph16193581

**Published:** 2019-09-25

**Authors:** Maria Teresa Montagna, Serafina Rutigliano, Paolo Trerotoli, Christian Napoli, Francesca Apollonio, Alessandro D’Amico, Osvalda De Giglio, Giusy Diella, Marco Lopuzzo, Angelo Marzella, Simona Mascipinto, Chrysovalentinos Pousis, Roberto Albertini, Cesira Pasquarella, Daniela D’Alessandro, Gabriella Serio, Giuseppina Caggiano

**Affiliations:** 1Department of Biomedical Science and Human Oncology, University of Bari “Aldo Moro”, Piazza G. Cesare 11, 70124 Bari, Italy; mariateresa.montagna@uniba.it (M.T.M.); serafina.rutigliano@uniba.it (S.R.); paolo.trerotoli@uniba.it (P.T.); francesca.apo@libero.it (F.A.); alessandro.damico@uniroma1.it (A.D.); osvalda.degiglio@uniba.it (O.D.G.); giusy.diella@uniba.it (G.D.); marcolopuzzo@gmail.com (M.L.); marzella.angelo@libero.it (A.M.); simona.mascipinto@uniba.it (S.M.); vpousis@gmail.com (C.P.); gabriella.serio@uniba.it (G.S.); 2Department of Medical and Surgical Sciences and Translational Medicine, “Sapienza” University of Rome, 0189 Rome, Italy; christian.napoli@uniroma1.it; 3Department of Civil, Building and Environmental Engineering, “Sapienza” University of Rome, 00184 Rome, Italy; daniela.dalessandro@uniroma1.it; 4Department of Medicine and Surgery, University of Parma, 43125 Parma, Italy; roberto.albertini@unipr.it (R.A.); cesiraisabellamaria.pasquarella@unipr.it (C.P.)

**Keywords:** operating theatre, orthopaedic surgery, surgical site infection, air quality

## Abstract

Postoperative infections are a concern, especially in total knee and total hip arthroplasty. We evaluated the air quality in orthopaedic operating theatres in southeastern Italy to determine the level of bacterial contamination as a risk factor for postoperative infection. Thirty-five hospitals with operating theatres focused on total knee and total hip arthroplasty participated. We sampled the air passively and actively before surgeries began for the day (at rest) and 15 min after the surgical incision (*in operation*). We evaluated bacterial counts, particle size, mixed vs turbulent airflow systems, the number of doors, number of door openings during procedures and number of people in the operating theatre. We found no bacterial contamination *at rest* for all sampling methods, and significantly different contamination levels *at rest* vs *in operation*. We found no association between the number of people in the surgical team and bacteria counts for both mixed and turbulent airflow systems, and low bacterial loads, even when doors were always open. Overall, the air quality sampling method and type of ventilation system did not affect air quality.

## 1. Introduction

Operating theatres (OTs) are particularly complex systems in which numerous risk factors favor the onset of infectious complications. These include the structural characteristics of the facility and its organization, the patient’s condition, the type of surgery, the behaviors of the surgical team, how often people enter and leave the OTs, and the efficiencies of the Heating, Ventilation and Air Conditioning (HVAC) systems [1,2,3,4,5].

Surgical site infection (SSI) is a devastating postoperative complication and has a substantial impact on morbidity and mortality. Patients with SSI have a 2–11 times higher risk of death than patients without SSI, and SSIs are associated with considerably higher costs, especially when associated with additional surgical procedures and a longer hospital stay [6,7,8].

In orthopaedic surgery, infection is the most common indication for revision in total knee arthroplasty and the third most common indication in total hip arthroplasty; when infection cannot be eradicated, treatment can include arthrodesis or even amputation [9]. In the USA, between 2001 and 2009, the risk of infection following hip and knee arthroplasties increased from 1.99% to 2.18% and from 2.05% to 2.18%, respectively [10]. By 2030, the risk is expected to increase to 6.5% and 6.8%, respectively [8,11]. In Italy, the most recent arthroplasty annual report underlined that 18.9% of knee arthroplasty revisions and 7.8% of hip revisions were secondary to infectious complications [12].

The air in OTs represents an important vehicle for microorganisms that cause SSI, especially in clean operations. After a Medical Research Council study showed a correlation between microbial air contamination and deep SSI incidence in prosthetic joint surgery, ultraclean OTs were recommended for this type of surgery [5]. Maximum air microbial contamination values were 10 colony-forming units per cubic metre (cfu/m^3^) when measured by active sampling and 350 cfu/m^2^/h or 2 cfu/9-cm plate/h (IMA, index of microbial air contamination) when measured by passive sampling [13]. Based on a meta-analysis by Bishoff et al. [14], the World Health Organization (WHO) guidelines recommend against unidirectional airflow ventilation systems to reduce the risk of SSI for patients undergoing total arthroplasty surgery. However, several criticisms exist regarding the studies included in the meta-analysis; in particular, none of the studies included an assessment of microbial air contamination and assumed that the mere presence of a unidirectional airflow system guarantees its proper function. Several factors can undermine the potential benefits of this ventilation system, in particular, incorrect behaviour of the surgical team. The “Infezioni del Sito Chirurgico in Interventi di Artroprotesi” (ISChIA) study evaluating hip and knee arthroplasties showed that more than half of the passive samplings during surgical activity in unidirectional airflow ventilation OTs had values higher than 2 cfu/9-cm plate/h, challenging the belief that unidirectional systems always provide acceptable airborne bacterial counts [15]. The same study showed that in turbulent airflow OTs (T-OTs), values much lower than the current recommended values of 180 cfu/m^3^ [16,17] and 25 IMA [18] can be obtained.

The role of airborne particles as a source of contamination has always been controversial, especially in clean surgical procedures [19]. Interpreting particle counts is based on the theory that particulate matter is considered a possible vehicle for microorganisms capable of contaminating surgical fields. The relationship between particle concentrations and microbial contamination is not yet fully accepted [20], although a recent study showed a strong correlation between air particle counts and microbial contamination, confirming that particle counts can be used for routine assessment of contamination in OTs [21].

Our study is part of a larger study, the IMPACT (IMproving the health of PAtients supporting dynamiC healTh systems and new technologies) project, promoted by the Apulia Regional Government (Southern Italy). The first part of this project, coordinated by the Department of Biomedical Sciences and Human Oncology of the University of Bari Aldo Moro (Apulia, Italy), was performed in collaboration with the Department of Civil, Construction and Environmental Engineering of Sapienza University of Rome, through a multidisciplinary and transversal approach. The first part of the study aimed to investigate whether OT design (shape, dimensions, layout and construction technology) influenced microbial air pollution [22].

The main objectives of the second part of this project were:to determine the degree of microbial air contamination at a defined time (1-day study) in empty *at rest*) and working (*in operation*) OTsto assess air quality in the OTs by comparing different sampling systemsto evaluate the association between microbiological data and particle counts with different HVAC systems.

## 2. Materials and Methods

### 2.1. Study Design

The IMPACT project was performed from 2015–2017. Initially, a census of all of the public hospitals in the Apulia Region (southeastern Italy) was performed to select hospitals equipped with orthopaedic OTs, with a special focus on those devoted to prosthetic surgery. The medical management staff of each hospital was contacted by email to request their participation in the study on a voluntary basis and without remuneration. Ethics approval was not necessary because the study was performed on environmental samples without involving human samples or data (Legislative Decree n. 196 of 30 June 2003).

Forty-five hospitals were identified in different areas of Apulia. Of these, 39 (86.7%) had operating areas, for a total of 287 OTs, and 35/39 (89.7%) had 49 orthopaedic OTs. Thirty hospitals with 35 orthopaedic OTs focused on hip and knee arthroplasty operations agreed to participate in the study (Figure 1). We performed air sampling using different sampling tools, to evaluate microbial and particle diffusion.

### 2.2. Air Sampling

For each OT, indoor air was sampled both *at rest* and *in operation* during the first surgical procedure (hip and knee arthroplasty) of the day as follows:(a)*at rest,* 1 h before the beginning of surgical activity, at the foot of the operative bed, to verify the efficiency of environmental cleaning systems and conditioner systems. All instruments started automatically 20 min after the technical team left the operating theatre.(b)*in operation,* 15 min after the surgical incision, at the foot of the operative bed, to verify the human impact on environmental pollution. During each operation, we collected detailed information concerning the number of staff in the OTs and the number of door openings.

Microbial contamination was evaluated through active sampling to measure the concentration of microorganisms in the air, and by passive sampling to measure the rate at which viable particles settled on surfaces [23].

The active sampling method was performed on both solid (Surface Air System, SAS Super ISO 180; PBI International, Milan, Italy) and liquid substrates (Coriolis^®^μ; Bertin Technologies, Montigny le Bretonneux, France). For each sampling, 1000 L of air was aspirated by SAS and Coriolis^®^μ, which were placed approximately 1 m above the floor and 1 m from the operating bed.

For SAS, the air microbial load was evaluated using 55-mm plates containing plate count agar (PCA; Becton–Dickinson, Heidelberg, Germany). After incubation at 36 ± 1 °C for 48 h, the number of cfu was adjusted using the conversion table provided by the manufacturer and was expressed in as cfu/m^3^.

For Coriolis^®^μ, a cone containing 15 mL of liquid substrate (0.005% Triton X-100) was used according to the manufacturer’s recommendations. After aspiration, the volume of liquid substrate was aeseptically transferred to a sterile container for culture-based investigations, and 0.5 mL of the original sampling solution and 1:10 dilutions were plated on PCA plates. The plates were incubated at 36 ± 1 °C for 48 h, and the average cfu was used to calculate the total airborne microbial load using the following equation [24,25]:(1)$$\mathrm{cfu}/m^{3}\;=\;\frac{\mathrm{cfu}}{\mathrm{Vplated}\;\mathrm{aliquot}\;\left[\mathrm{mL}\right]}\;\times \;\frac{\mathrm{dilution}\;\mathrm{factor}\;\times \;\mathrm{Vbuffer}\;\mathrm{after}\;\mathrm{sampling}\;\left[\mathrm{mL}\right]}{\mathrm{Vair}\;\mathrm{sample}\;\left[m^{3}\right]}$$

Passive sampling was performed using two sedimentation plates, 90-mm in diameter, which were exposed to the air for 1 h, 1 m above the floor on a Sed-3Unit^®^ (MRC AG, CH-9450 Altstätten/SG) at least 1 m away from major obstacles. The results were expressed as the mean of the two plates to determine the IMA. Total viable count (TVC) was recorded in duplicate to ensure sampling accuracy, using PCA plates that we incubated at 36 ± 1 °C for 48 h [26].

### 2.3. Particle Counts

Airborne particles with a diameter ≥ 0.5 μm and ≥ 5 μm in size were counted with a laser particle counter (Climet CI 754; Rigel Srl, Rome, Italy), which was certified and validated in accordance with the ISO 21501-4:2007 [27] requirements. The suction volume was 75 L/min, and measurements were performed three times, 1 m above the floor, and were expressed as the number of particles per m^3^. We used a 10-s interval between each of the three samplings. Figure 2 shows the floor plan of a representative OT and the positions of the air and particulate samplers. During the *in operation* samplings, we placed the instruments at the end of the plenum to avoid interfering with the surgical activities.

### 2.4. Statistical Analyses

Continuous variables did not approach Gaussian distribution; therefore, they were reported as the median and interquartile range (IQR). Comparisons between groups were performed with nonparametric tests: Wilcoxon’s and Kruskal–Wallis tests for independent groups, Wilcoxon’s and Friedman’s tests for nonindependent groups. Post-hoc comparisons were performed with the Conover test for multiple comparisons.

The relationships between variables were evaluated with Spearman’s correlation coefficient; a partial Spearman’s correlation was used to evaluate coefficients adjusted for other variables, such as particle size, number of people in the OT, number of door openings and the number of people involved in the surgery team.

Qualitative variables were analysed using the chi-square test or Fisher’s exact test, as appropriate. The number of door openings was classified as: “always open” (i.e., constantly open) or “occasionally open” (indicating that doors were open only when a health worker came in or out of the OT). To evaluate the relationships between air quality and door opening, microbial counts were classified as 0 cfu/m^3^ or >0 cfu/m^3^, and we defined two grades for passive sampling counts: 0 (zero) IMA and >0 (zero) IMA.

A *p*-value < 0.05 indicated statistical significance. Analyses were performed using the statistical software MedCalc 18.6 (MedCalc Software, Mariakerke, Belgium) and SAS 9.4 (SAS Institute Inc., Cary, NC, USA).

## 3. Results

Among the 45 hospitals present in Apulia from 2015–2016, 35 orthopaedic OTs were part of a complex that provided adequate space for anaesthesia and surgery and were under positive pressure regarding the adjacent areas (≥ 5 Pa). The 35 orthopaedic OT systems with air treatment were equally distributed: 18 OTs were equipped with a T-OT and 17 with an M-OT. No unidirectional OTs were included in this study.

For each OT, the median number of doors was two for both M-OTs and T-OTs, and the door type (sliding or swinging) was similar. The doors were kept “always open” in 58.8% of M-OTs and in 55.6% of T-OTs. Regarding the “occasionally open” doors, the median number of openings for swinging doors was 16 in M-OTs and 17 in T-OTs, and the median number of openings for sliding doors was 11.5 in M-OTs and 16 in T-OTs. Table 1 shows the characteristics of M-OTs and T-OTs.

The median number of people present in the OTs *in operation* was nine in both types of OT, with no statistically significant difference between M-OTs and T-OTs (*p* > 0.05).

Regarding microbiological data, 280 air samples (140 *at rest* and 140 *in operation*) were studied from the 35 OTs. With all systems used, the median value indicating bacterial contamination *at rest* was zero for both M-OTs and T-OTs. *In operation*, the median value with the SAS system was 15 cfu/m^3^ (M-OTs) and 23.5 (T-OTs) cfu/m^3^ (IQR: 7.5–60 and 17–58, respectively). With the Coriolis^®^μ system, the median values were 48 cfu/m^3^ in M-OTs and 10.5 cfu/m^3^ in T-OTs (IQR: 24–67.75 and 0–52, respectively); with the sedimentation plates, IMA values were 4 and 4.5 (range: 2.75–6 and 4–8, respectively) (Table 2).

A statistically significant increase in bacterial contamination was observed when comparing samples *at rest* vs *in operation* for both M-OTs and T-OTs and with all methods.

Regarding particle data, there were no statistically significant differences in the 0.5 µm particle counts between M-OTs and T-OTs *at rest* or *in operation*. Table 3 shows the partial Spearman’s correlation coefficient between microbial counts *in operation* and for each of the following variables: particles, number of access doors in the OT and number of people present in the OT.

Statistically significant correlation were obtained for the bacterial count regarding Coriolis^®^μ and particles ≥ 5 µm (r_S_ = 0.68, *p* = 0.0158), in M-OTs *in operation*. These results show a direct correlation between particle counts using Coriolis^®^μ and ≥ 5 µm particles; the bacterial count and particles were concordantly high.

M-OTs had a median number of 1 (range 1−4) sliding doors, and 41.2% were always open; in 10 OTs the median number of openings was 10 (range 5–31). T-OTs had a median number of 2 sliding doors (range 0–3), and 44.4% were always open. There was no statistically significant difference in the number of door openings between T-OTs and M-OTs (*p* = 0.5401). Table 4 shows the percentages of OTs by door openings and class of bacterial counts, stratified by airflow and sampling systems. There was no statistically significant relationship between door opening and bacterial count by any method or in M-OTs and T-OTs. This association was not statistically significant for mixed airflow (*p* = 0.134769) or turbulent airflow (*p* = 1).

Regarding particle data, regression analysis for associations between bacterial counts and particle size showed a nonunivocal pattern (Figure 3a,b).

We deleted the number of people in the OTs, the number of doors (sliding and swinging) and the number of openings from any model when these variables resulted in no statistical significance in any model. The type of airflow was not statistically significant, but we left this variable in the models, to evaluate its effect.

*At rest*, the effect of particle size on Coriolis^®^μ counts could not be determined because 85% of OTs (30/35) had a count of zero.

## 4. Discussion

This study evaluated air quality in orthopaedic OTs, both *at rest* and *in operation* during the first surgical procedure of the day, using different sampling systems and data for different types of HVAC systems. The difference between TVC mean values in M-OTs and T-OTs was not statistically significant. We used both active and passive sampling methods [17,26] and in addition to traditional sampling methods, we used the Coriolis^®^μ active sampler because some authors showed that Coriolis^®^μ had a higher sensitivity when the study focused on microorganisms that were difficult to isolate from air samples [28].

In our study, we found a statistically significant correlation between the microbial count obtained by Coriolis^®^μ and the particle count; M-OTs showed a higher microbial count if particles ≥ 5 μm were considered. Based on these results, we hypothesized that larger particles carry a greater number of microbes. However, these results were not confirmed when the SAS system was used, although this is also an active sampling method, similar to other studies showing different results when different samplers were used [29,30]. The role of airborne particles as a source of contamination, especially during clean surgical procedures, is still debated and also although other authors studied the problem of correlation between microbial count and particles, the question is yet unresolved [2,20,31,32].

No bacterial contamination appeared in OTs *at rest* with any sampler we used. Although other authors reported high contamination values before daily activities began in the OTs [33,34,35,36], our data suggest effective sanitation measures in the OTs in our study and correct functioning of the ventilation systems. According to a previous study [22], air quality *at rest* confirmed the effectiveness of the sanitation measures of the OTs and of the HVAC systems.

Higher bacterial counts appeared *in operation,* in our OTs, using both active and passive sampling methods. The median values were low, although differences in bacterial counts before and during surgery were significant for all systems. Indeed, the microbiological results obtained *in operation* by the SAS system showed median values of < 15 cfu/m^3^ in M-OTs and 23.5 cfu/m^3^ in T-OTs; i.e., lower than the limit of contamination recommended by ISPESL guidelines [17] in OTs *at rest* (< 35 cfu/m^3^). These results suggest that the current recommended value appears to be too high for conventional OTs. Most likely, in accordance with the ISPESL guidelines [17], only by performing repeated sampling over time and acquiring a historical database for each type of OT, would it be possible to calculate the alert values. Currently, air sampling methods are not standardized; therefore, it is difficult to compare results obtained from different samplers. The debate on the choice of these systems is still open.

Regarding the presence of people in the OTs as a factor influencing airborne contamination during surgical procedures, some authors indicated that the number of members in the surgical team had the greatest impact on both airborne bacteria and particle counts [15,32]. In our study, Spearman’s correlation coefficient analysis did not reveal a significant association between the number of people (median = 9) and the airborne bacterial count in both M-OTs and T-OTs. Even when the doors (sliding or swinging) were always open, the bacterial load was very low, probably because the bacterial load was equal to the adjacent area. The impact of door openings on air quality was analysed by other authors [37,38] but clinical confirmation remains complex. Some authors [39] found no significant differences in the airborne bacterial count between closed doors.

## 5. Conclusions

In conclusion, our results did not show significant differences between our measured variables, and initial observations support the hypothesis that air quality in OTs is not affected by sampling methods or different ventilation systems. *At rest,* our results confirmed the effectiveness of the HVAC in the OTs, i*n operation*, TVC values were within the recommended threshold. Further study will help assess other aspects of this study using a more detailed protocol and repeated measurements over time.

## Figures and Tables

**Figure 1 ijerph-16-03581-f001:**
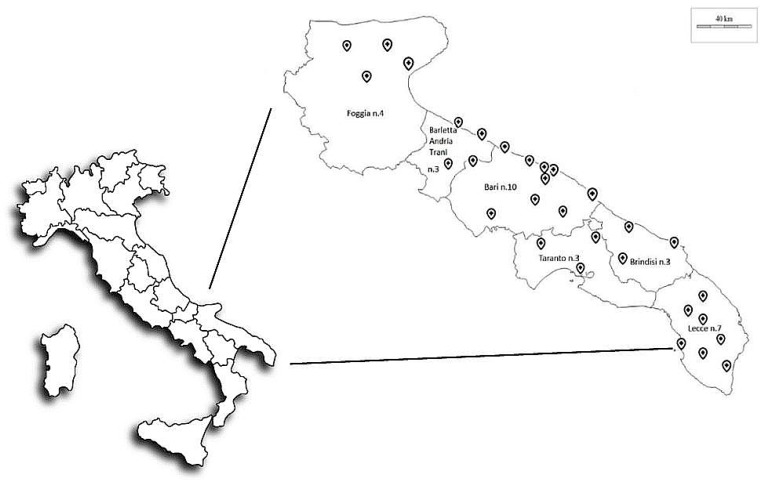
Location of the 30 enrolled hospitals in Apulia, Southern Italy.

**Figure 2 ijerph-16-03581-f002:**
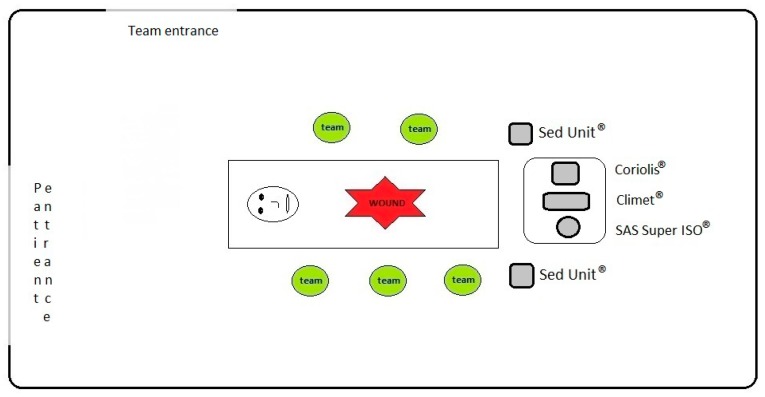
Position of the air sampling instruments in the operating theatres during the *in operation* samplings.

**Figure 3 ijerph-16-03581-f003:**
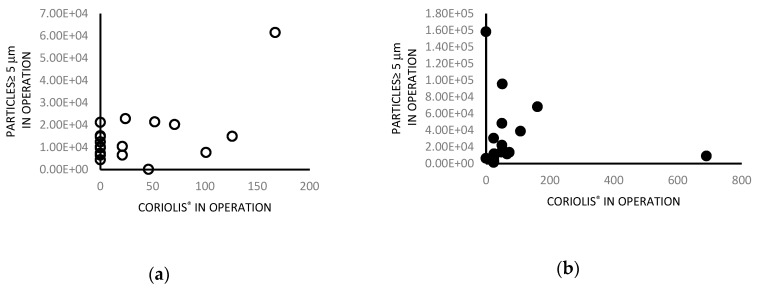
(**a**) Scatter plots evaluating the relationship between counts using the Coriolis^®^μ and particles ≥ 5 µm (r_S_ = 0.68, *p*-value = 0.0158) in operating theatres with mixed airflow. (**b**) Scatter plots evaluating the relationship between counts using the Coriolis^®^μ and ≥ 5 µm particles (r_S_ = 0.47, *p*-value = 0.1041) in operating theatres with turbulent airflow.

**Table 1 ijerph-16-03581-t001:** Comparison of the characteristics between the mixed airflow (M-OTs) and turbulent airflow operating theatres (T-OTs). IQR = Interquartile Range.

Variables	M-OTs (*N* = 17)		T-OTs (*N* = 18)		*p-*Value
	Median	IQR	Median	IQR	
**DOORS**	2	1–2	2	1–3	0.9721
Sliding doors	1	1–1	2	1–2	0.1697
Swinging doors	0	0–1	0	0–0	0.0899
Volume of OTs	114	105–123	140	114–170	0.0729
Surface of OTs	38	34–42	42	39–47.5	0.1922
Number of air changes/h	15	15–26	18.29	15–18.5	0.8162

**Table 2 ijerph-16-03581-t002:** Median values for bacterial contamination *at rest* and *in operation* obtained by active (SAS, Coriolis^®^μ) and passive (sedimentation plates) methods.

Method	At Rest				In Operation			
	Mixed		Turbulent		Mixed		Turbulent	
	Median	IQR	Median	IQR	Median	IQR	Median	IQR
SAS (cfu/m^3^)	0 ^$^	0–2	0.5 ^#^	0–3	15 ^$^	7.5–60	23.5 ^#^	17–58
Coriolis^®^μ (cfu/m^3^)	0 ^	0–0	0	0–0	48 ^	24–67.75	10.5	0–52
Settling plates (IMA)	0 *	0–1	0 ^§^	0–2	4 *	2.75–6	4.5 ^§^	4–8

^$, #, ^,^ *, ^§^: paired comparisons, statistically significance at *p* = 0.0023. SAS = Surface Air System; IMA = index microbial air contamination; IQR = interquartile range.

**Table 3 ijerph-16-03581-t003:** Partial Spearman’s correlation coefficient for comparisons between particle size, number of access doors to the operating theatres and the number of people during *in operation* measurements.

**Variables**	**Mixed Airflow**					
	**SAS**		**Coriolis^®^μ**		**Settling Plates**	
	**Spearman Coeff.**	***p*-Value**	**Spearman Coeff.**	***p*-Value**	**Spearman Coeff.**	***p*-Value**
**Particles ≥ 0.5 µm**	−0.071	0.818	0.045	0.8839	−0.125	0.6848
**Particles ≥ 5 µm**	0.015	0.9631	0.68	0.0158	0.006	0.9854
**Number of doors**	0.048	0.8822	0.32	0.3147	−0.063	0.8452
**Number of people**	0.44	0.1496	−0.15	0.6489	0.26	0.4228
**Variables**	**Turbulent Airflow**					
	**SAS**		**Coriolis^®^μ**		**Settling plates**	
	**Spearman Coeff.**	***p*-Value**	**Spearman Coeff.**	***p*-Value**	**Spearman Coeff.**	***p*-Value**
**Particles ≥ 0.5 µm**	0.24	0.4542	0.62	0.0316	0.23	0.4786
** ≥ 5 µm**	0.47	0.1015	0.47	0.1041	0.16	0.5988
**Number of doors**	−0.12	0.6972	−0.27	0.3666	0.31	0.3002
**Number of people**	0.17	0.5775	0.34	0.2542	0.34	0.4315

SAS = Surface Air System.

**Table 4 ijerph-16-03581-t004:** The number of door openings and bacterial counts stratified by airflow and sampling system.

**Mixed Airflow**					
	**Door Openings (OTs = 10)**		**Doors Kept Open (OTs = 7)**		***p*-Value**
	**No.**	**%**	**No.**	**%**	
**SETTLING PLATES (IMA/plate)**					
**0**	rs	rs	0	0.0	1
**>0**	10	100.0	7	100.0	
**SAS (cfu/m^3^)**					
**0**	0	0.0	0	0.0	0.134769
**>0**	10	100.0	7	100.0	
**Coriolis^®^μ (cfu/m^3^)**					
**0**	0	0.0	2	28.6	0.1544
**>0**	10	100.0	5	71.4	
**Turbulent Airflow**					
	**Door Openings (OTs = 10)**		**Doors Kept Open (OTs = 7)**		***p*-Value**
	**No.**	**%**	**No.**	**%**	
**SETTLING PLATES (IMA/plate)**					
**0**	0	0.0	1	12.5	0.44444
**>0**	10	100.0	7	87.5	
**SAS (cfu/m^3^)**					
**0**	0	0.0	0	0.0	1
**>0**	10	100.0	8	100.0	
**Coriolis^®^μ (cfu/m^3^)**					
**0**	6	60.0	3	37.5	0.6372
**>0**	4	40.0	5	62.5	

OTs = operating theatres; IMA = index microbial air contamination; cfu = colony-forming unit; SAS = Surface Air System.

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
