# Peer review of "Evaluation of Air Contamination in Orthopaedic Operating Theatres in Hospitals in Southern Italy: The IMPACT Project"

_ijerph, 2019, doi:10.3390/ijerph16193581_

Round 1

Reviewer 1 Report

The paper is interesting and the attention of the authors for a medico-legal issue is valuable. The paper is well written and methodological approach rigorous. 

Author Response

The authors thank for the comment and appreciation of the manuscript.

Reviewer 2 Report

Brief summary

The authors report a study with the main goal of determine evaluate the air quality in orthopaedic operating theatres (OTs) using different sampling systems and data from different types of heating, ventilation and air conditioning (HVAC). The characteristics between the mixed- and turbulent air flow operating theatres was evaluated as well as the bacterial contamination at rest and in operation OT obtained by active and passive methods. Additionally, the effect of access doors and number of people during in operation measurements was also considered. The authors conclude that no significant differences between measured variables in operation compared with at rest, indicative of that air quality is not affected by sampling methods or different ventilation systems and reinforce the focus on sanitation measures of the OTs and of the HVAC systems as preventive measure to orthopaedic postoperative infections.

Broad comments

It is a well design study in a relevant topic considering the need of increase the knowledge about the air quality in orthopaedic operating theatres and the controversial studies about this thematic.

The manuscript is well organized, containing all the components expected. All the sections are well-developed. The article is well-written and is easy to understand.

The Abstract is concise and adequately summarizes the article content.

The Introduction represents an adequate synthesis of the literature. However it is mentioned the main objectives of the second part of project IMPACT (lines 46-51) but the questions sets out to answer and the main goal of the study should be clearly mentioned herein.

The Materials and Methods are mostly clearly explained. The 4 Tables presented are useful to understand the results obtained.

The figures are presented with a lack of image quality and should be improved, especially the Figure 3.

The mention to the previous study (line 50) should be made at Discussion and not at Conclusion. Additionally, if the sanitation measures are more critical than the sampling methods or different ventilation systems it should be discussed.

Author Response

The authors thank for the comments and for the valuable suggestions, so  in accordance with the Reviewer:

1) they clarify the objectives of the study "lines 46-51"

2) the quality of all figures and in particular figure 3 has been improved;

3) they modified the conclusions "line 50"

Reviewer 3 Report

The article is interesting, but I think some things should be clarified:

1.-the authors have taken air samples at rest and 15 minutes after surgical incision. The authors should explain the number of surgical interventions performed on the same day at one orthopedic operating theater. Is it possible that on the same day, more than one patient had surgery on the same OT? In this case, the authors should clarify if the sampling was performed 15 minutes after surgical incision for each of the surgical interventions. Were there differences in air pollution depending on whether the surgical intervention was the first or not?

2.-The article tries to determine the level of contamination as a risk factor for surgical infection, but this issue is very controversial. Therefore, the authors should clarify in the discussion the real usefulness of the study and why the authors believe their research is relevant.

3.-The authors find that M-OTs showed a higher microbial count if particles> 5 um were considered. Have other authors found a relationship between particle size and microbe count? The authors should clarify this point.

Author Response

The authors thank the Reviewer for his precious observations, and they clarify the points required:

1) the study protocol provided the sampling of air in OTs only for the first surgical      intervention of the day, so as to be able to compare the sampling OTs only for the first surgical      intervention of the day, so as to be able to compare the sampling results obtained in all orthopedic operating rooms of the different   hospitals.

2) Regarding to the study subject, since the literature data are very controversial, this manuscript aims to record, in the same time, the level of microbial and particle contamination in a large geographical area of Southern Italy. Moreover, the use of the same tools used by the same operators in different OTs of  hospitals located in areas distant from each other, allowed to better standardize the study protocol. OTs of  hospitals located in areas distant from each other, allowed to better standardize the study protocol. 

3) other studies have evaluated the correlation between particles and microbial count, bringing controversial results and leaving the issue unresolved. The author clarify this point  in the manuscript.

Round 2

Reviewer 3 Report

The authors have responded to the problems raised. On the other hand, they have made small changes in the discussion that improve the understanding of the importance of the present study.